# Preparedness of tertiary care hospitals to implement the national TB infection prevention and control guidelines in Bangladesh: A qualitative exploration

Md. Saiful Islam [1,2¤]*, Sayeeda Tarannum[1], Sayera Banu[1], Kamal Ibne Amin Chowdhury [1], Arifa Nazneen[1], Abrar Ahmad Chughtai[2], Holly Seale[2]

1 Infectious Diseases Division, Program for Emerging Infections, icddr,b, Dhaka, Bangladesh, 2 School of Population Health, Faculty of Medicine and Health, University of New South Wales, Sydney, Australia

¤ Current address: School of Population Health, University of New South Wales, Sydney, Australia
* saiful@icddrb.org

## Abstract

In high tuberculosis (TB) burden countries, health settings, including non-designated TB hospitals, host many patients with pulmonary TB. Bangladesh's National TB Control Program aims to strengthen TB infection prevention and control (IPC) in health settings. However, there has been no published literature to date that assessed the preparedness of hospitals to comply with the recommendations. To address this gap, our study examined healthcare workers knowledge and attitudes towards TB IPC guidelines and their perceptions regarding the hospitals' preparedness in Bangladesh. Between January to December 2019, we conducted 16 key-informant interviews and four focus group discussions with healthcare workers from two public tertiary care hospitals. In addition, we undertook a review of 13 documents [i.e., hospital policy, annual report, staff list, published manuscript]. Our findings showed that healthcare workers acknowledged the TB risk and were willing to implement the TB IPC measures but identified key barriers impacting implementation. Gaps were identified in: policy (no TB policy or guidelines in the hospital), health systems (healthcare workers were unaware of the guidelines, lack of TB IPC program, training and education, absence of healthcare-associated TB infection surveillance, low priority of TB IPC, no TB IPC monitoring and feedback, high patient load and bed occupancy, and limited supply of IPC resources) and behavioural factors (risk perception, compliance, and self and social stigma). The additional service-level gap was the lack of electronic medical record systems. These findings highlighted that while there is a demand amongst healthcare workers to implement TB IPC measures, the public tertiary care hospitals have got key issues to address. Therefore, the National TB Control Program may consider these gaps, provide TB IPC guidelines to these hospitals, assist them in developing hospital-level IPC manual, provide training, and coordinate with the ministry of health to allocate separate budget, staffing, and IPC resources to implement the control measures successfully.

**Data Availability Statement:** All relevant data are within the manuscript and Supporting information files.

**Funding:** This research protocol was funded by the United States Centre for Disease Control and Prevention (CDC) through the cooperative agreement grant number 5U01GH1207. MSI and SB received this award. None of the authors received any other funding for this research. The funders had no role in study design, data collection, and analysis, decision to publish, or preparation of the manuscript.

**Competing interests:** All authors report no conflicts of interest relevant to this article.

# Introduction

Healthcare workers (HCW) and other hospital occupants are at increased risk of tuberculosis (TB) infection in high TB burden countries [1]. In Bangladesh, TB specialized hospitals admit and treat patients with drug-susceptible and drug-resistant TB as inpatients. Prior studies show that TB specialized hospitals implement TB IPC guidelines regularly [2]. On the other hand, tertiary care general hospitals admit presumptive TB patients for diagnosis and occasionally provide treatment to TB patients with comorbidities. Prior studies that pulmonary TB patients stay on average 5.5 days. There was the minimal implementation of TB IPC measures in public tertiary care general hospitals [3]. In addition, pulmonary TB patients in inpatients wards with limited or no IPC measures have been shown to increase HCW exposure to TB, resulting in 42% of latent TB infections among HCW [4].

TB infection prevention and control (IPC) is one of the World Health Organization's (WHO) 12 collaborative activities to prevent TB [5]. In low-and middle-income countries (LMIC), TB IPC is considered a critical strategy for TB prevention and control in health settings [6]. International TB IPC guidelines that target health settings can be traced back to the 1980s, and the most recent TB IPC guidelines was updated and published in 2019 by the WHO [7, 8]. As part of the WHO's TB control strategy, many LMIC have developed national TB IPC guidelines; however, the implementation of these guidelines remains inconsistent due to multifaced factors prevailing in diverse health settings [8]. Bangladesh developed its national TB IPC guidelines in 2011, which includes a three-level hierarchy of control measures: administrative, environmental, and personal respiratory protection [9]. In 2019, Bangladesh ranked seventh among the 30 high TB burden countries globally [10]. The estimated incidence rate of all forms of TB per 100,000 population was 221 [10].

Bangladesh's National TB control program (NTP) is working to strengthen TB IPC in health settings. However, there has been no published literature to date that assessed the preparedness of hospitals to comply with the recommendations. To support NTP, it is crucial to understand the local context and broader health system factors, including behavioural and organizational factors [11]. An assessment of target facilities to understand the barriers, leadership presence, attitudes of frontline HCW, and local leaders in TB IPC programs can support the tailoring and implementing guidelines [12, 13]. We adopted the WHO IPC Assessment Framework (IPCAF) tool to conduct the hospital preparedness assessments for TB IPC [14]. The assessment aimed to examine HCW' knowledge and attitudes towards TB IPC guidelines and the preparedness of two public tertiary care hospitals in Bangladesh to implement the guidelines.

# Materials and methods

## Study sites and participants

Between January and December 2019, a qualitative study was undertaken involving two public tertiary care hospitals: Hospital A and Hospital B. Key characteristics of the hospitals have been presented in Table 1. The rationale for selecting these hospitals was that they admitted approximately 400 adult pulmonary TB patients annually. Staff members involved in TB patient management were recruited from: medicine wards, directly observed therapy short-course (DOTS) clinics, radiology department, and administrative units. To recruit participants for interviews and focus group discussions (FGD), the team made a list of HCW who were either members of hospital committees, or those who were deemed to be influential in decision-making, knowledgeable, and experienced in dealing with infectious disease patients, or those staff who were seen as influence other's practices. The team then selected those with the highest duration working in tertiary care hospitals.

**Table 1. Key characteristics of the study hospitals, 2019.**

| Characteristics | Hospital A | Hospital B |
|---|---|---|
| Location | • Located in Barisal. | • Located in Rajshahi. |
| | • 123 kilometres away from Dhaka, the capital city of Bangladesh | • 198 kilometres away from Dhaka |
| Number of beds | • 1000 bed-hospital | • 1200-bed hospital |
| Number of patients (approximate) receive treatment | • 406,400 as outpatients | • 806,541 as outpatients |
| | • 117,263 as emergency patients | • 12,154 as emergency patients |
| | • 121,891 as in-patient | • 175,868 as inpatients |
| Number of Doctors | • 323 | • 395 |
| Number of Nurses | • 768 | • 1145 |
| Number of ancillary care workers | • 392 | • 514 |
| Bed occupancy rate | • In 2018, the bed-occupancy rate was 155.8% | • The bed occupancy rate in hospital B was 151.5%. |
| DOTS clinic | • Present within the hospital premises | • Present within the hospital premises |
| | • Provide diagnosis and treatment support to TB patients between 8:30 AM to 2 PM during the weekdays | • Provide diagnosis and treatment support to TB patients between 8:30 AM to 2 PM during the weekdays |
| X-ray | • Inside the hospital facilities that runs 24-hours a day | • Inside the hospital facilities |
| GeneXpert | • GeneXpert facilities available in the neighbouring TB specialty hospitals | • GeneXpert facilities available in the neighbouring TB specialty hospitals |
| Surveillance for healthcare-associated TB infection | • Not present | • Not present |
| TB IPC committee | • Not present | • Not present |
| TB IPC guidelines | • Not present | • Not present |
| General IPC guidelines | • Not present | • Not present |
| TB IPC training | • Not present | • Not present |
| Cough screening | • Not present | • Not present |
| Separation of patients with pulmonary TB | • Partially present | • Partially present |
| | • TB patients with known pulmonary TB occasionally separated in the corner of the inpatient ward | • TB patients with known pulmonary TB occasionally separated in the corner of the inpatient ward |
| Isolation | • No isolation room was available for TB patients. | • No isolation room was available for TB patients. |
| Ventilation | • Natural ventilation | • Natural ventilation |
| | • Ceiling fans available | • Ceiling fans available |
| Visitor control | • There is a policy on visiting hours. | • There is a policy on visiting hours. |
| | • No implementation of visitor control | • No implementation of visitor control |
| Supply of PPE | • A limited supply of IPC resources | • A limited supply of IPC resources |
| | • No TB specific IPC supplies | • No TB specific IPC supplies |
| | • No supply of N95 resources | • No supply of N95 resources |

## Data collection

In order to get a rich understanding of the existing IPC practices, hospital infrastructure, and resources available for TB IPC, as well as explore the organizational preparedness for change, the study included a hospital documents review, key-informant interviews (KII), and focus group discussions (FGDs). The use of multiple data sources allowed us to triangulate the data. The KII and FGD guidelines were developed based on the IPCAF tool published by WHO in 2018 to assist the implementation of the WHO Core Components of IPC programs in health settings [15, 16]. The use of this tool in the guidelines helped us to detect relevant gaps in hospital preparedness. A field team of three social scientists trained in qualitative research was employed to collect the data. The team has several years of experience working in hospital IPC. Before data collection, the team received a week-long training on data collection tools,

participant recruitment, interacting with participants, and training on how to be reflexive, reflective, and minimize subjectivity during interviews and FGDs.

The data collection team had more than five years of work relationship with the study participants for which they have a good rapport with the participants of interest to ensure trust. Due to prior work experience in the study settings, the data collection team was aware of the hospitals' context and social settings, which allowed the team to ensure good reflexivity. Methodological triangulation is another measure of rigour in qualitative research. The use of KIIs, FGDs, and document reviews confirmed the internal validity of the data collected under the study. To ensure the interpretative rigor of this research, we presented the data and the analysis with study participants to cross-check and ensure that all their responses were reflected well in the study.

**Key-informant interviews.**   Through KII, the team explored (i) hospital infrastructure, context, and readiness, (ii) HCW risk perception, concerns, attitudes and anticipated barriers regarding the TB IPC implementation, and (ii) the role that HCW (and others) can play in overcoming the barriers. The interviews were conducted in HCW' office room. The average duration of interviews was 49 minutes, and the interviews were audio-recorded. However, two key-informants refused to be recorded in the interviews, and then detailed hand notes were taken.

**Focus group discussions.**   The FGDs were conducted in a hospital conference/meeting room where six to 10 people participated in each. Two team members conducted the focus groups: one moderated the session, and the other took detailed notes. Like KII, FGDs were also audio recorded. We convened the FGDs to elicit HCW perspectives on more complex phenomena (e.g., power relation, hospital priority, social and cultural aspects of TB, and TB among healthcare workers). The conversational nature during the FGDs allowed participants to share their perspectives and listen to and reflect on other participants perspectives. The sessions ranged from 1hour 30 minutes to 2 hours.

**Document review.**   TB IPC policies and historical documents were reviewed to understand gaps in the policy contents, hospital characteristics, building design, in-patient-ward layout and renovation, bed capacity, and bed-occupancy rate. The reviewed documents included national TB IPC policy, national and hospital level health bulletins, published and unpublished reports; scientific manuscripts focused on study hospitals; and hospital staff lists.

## Data analysis

All the data were collected in Bengali. We performed a descriptive analysis of the document review data. The team transcribed the audio-recorded interviews and FGDs verbatim and typed them into word. The audio recordings were repeatedly listened to, compared with the transcriptions to check for accuracy. Initially, two authors reviewed the field notes, two KIIs, and one FGD and developed a code list. The coding system was based on a literature review, study objectives, themes, and sub-themes related to TB IPC measures and emerging themes [17]. The transcribed data and field notes were transferred into text organising software (NVivo) to facilitate coding and analysis. The lead author then read the interviews and FGDs and coded the excerpts of texts according to the code list. When new information could not be coded with the existing code list, the author added new codes, and the process continued until the end. The intercoder reliability was achieved through discussions among the two coders about code definitions and emerging themes [18]. The coded data were then categorised under different themes that emerged, and according to each theme, a summary was developed in English [17]. All the authors reviewed the themes and sub-themes for consensus and reliability. Our analytical framework included three categories consisting of 10 dimensions- (a) policy

level factors: i) TB IPC guidelines; (b) health system factors: ii) TB IPC programs; iii) education and training; iv) HAI surveillance; v) Multimodal strategies including system and culture change; vi) monitoring and evaluation; vii) workload, staffing, and bed occupancy; viii) the built environment, materials and equipment, and (c) behavioural level factors: ix) Risk perception; x) compliance with personal protective equipment; and xi) Stigma.

**Study definitions.** Adopting from Weiner, we defined organisational readiness as the extent to which organisations are prepared infrastructurally, administratively, and with resources, and HCW are prepared psychologically, behaviorally and willing to accept the changes due to TB IPC implementation [19].

## Ethical approval

The authors obtained written informed consent from all the participants before the KIIs and FGDs. The study protocol was reviewed and approved by UNSW ethical review committee (HC180517) and icddr,b institutional review board (PR 16090).

## Results

The team conducted 16 KIIs and four FGDs. The field team approached 18 HCWs for KIIs and 38 HCWs for FGDs, where 16 HCW consented to participate as key informants and 35 consented as FGD participants. Among all participants, 29 were doctors, 20 were nurses, and two were ancillary workers. The participants' mean age was 39 years (standard deviation (SD)-9.6), and 32 were male. The key-informants had an average of 20.6 years (SD:10.8) of work experience, and the FGD participants had an average of 6.8 years (SD:6.3) of work experience in public tertiary care hospitals. We report policy level, hospital level, and behavioural level factors that may impact the implementation of TB IPC measures.

## Policy level factors

**TB IPC guidelines.** The policy review found gaps in the content of the guidelines and many of the guideline recommendations were not context-appropriate. For example, the national TB IPC guidelines recommend triage and isolation of presumptive TB patients. However, none of the hospitals had dedicated TB wards or isolation rooms where presumptive TB patients could be isolated. Besides, no TB IPC guidelines were available in the study hospitals. Almost all the key informants and FGD participants said they were unaware of national TB IPC guidelines. One of the key informants mentioned that he had been working in the facility since 1957 and has never seen a copy of the TB IPC guidelines. The participants added that they did know about the standard IPC measures.

## Hospital level factors

**TB IPC programs.** All the participants said that the study hospitals lacked a TB IPC program with a dedicated IPC lead, focal person, committee, or coordination body. The participants reportedly did not receive any demonstrable support from NTP for TB IPC activities. One of the participants from an FGD said,

> To my knowledge, there is no TB IPC committee in this hospital. There might be a committee on pen and paper, but I did not hear about this or did not see the committee's activity. There was no publicity about any TB IPC committee in the hospital. If there was a TB IPC committee, I must be involved there.

There was no separate budget for the TB IPC program in the hospital. FGD participants mentioned that they received funding according to the number of beds in the hospital, not according to the number and types of patients they admit and treat.

**TB IPC education and training.** None of the study participants reported that they received any dedicated training for TB IPC. One key informant reasoned that since the study hospitals were not TB specialized hospitals and TB IPC was not a priority, they did not receive training. Some doctors mentioned that they received education and training on TB patient treatment and management, and there were brief discussions on TB IPC. One FGD participant mentioned that all levels of HCW did not have access to any training. Only a few people, particularly the doctors, received training repeatedly. The nurses mentioned that they often provide health education, including cough etiquette and respiratory hygiene, to patients and family caregivers, but no TB IPC training was offered to patients.

The doctors in FGD highlighted the necessity of TB IPC training as-

*We need training on TB IPC. Since the TB IPC strategies may update over time, we need refresher training every year. The authority may train a few senior HCW on TB IPC at the national level. These trained HCW can train others in the hospital at the local level. We need to follow the training for trainers (ToT) approach. Following this approach, the trainers will learn more and be efficient when providing training to other HCW.*

**Healthcare-associated TB infection surveillance.** All the participants mentioned that there was no surveillance for healthcare-associated TB infection in the hospitals. They mentioned that surveillance would require financial and human resources, and the hospital lacked TB-specific resources. Moreover, they added that they never received any instruction from the health ministry to run surveillance on healthcare-associated TB infection.

**Multimodal strategies including system and culture change.** The key informants mentioned that public tertiary care general hospital served a variety of patients. Therefore, these hospitals have diverse priorities. Two of the key informants said that the primary focus of the hospitals was patient investigation and treatment. Therefore, TB IPC remained a neglected part. One of the KII participants who was the head of medicine wards mentioned-

*This is a public tertiary care teaching hospital. We do not only treat TB patients, but we also admit all kinds of patients. It is not possible to put so much effort into TB patients. TB screening and isolation on admission is possible only in TB specialty hospitals, not here (tertiary care general hospital). We receive 1 percent or 1.5 percent of the total patients with TB. We will not be able to implement TB IPC recommendations for this small proportion of TB cases. It (TB IPC) is not possible to implement here.*

The nurses from one FGD informed us that they do not have easy access to discuss IPC with the hospital director. One of the FGD participants shared about the chain of information sharing as-

*(For any infection control issue) We need to talk to the assistant registers at the inpatient wards first, and then the assistant registers talk to the registers. The registers then talk to the professors, and the professors then discuss the issue with the hospital director.*

**Monitoring and feedback.** The participants informed us that the hospitals lacked a well-defined and targeted TB IPC monitoring and feedback plan. The was no evaluation of patient and HCW safety culture in the hospitals. One of the key informants mentioned that there

might be a monitoring and feedback culture for clinical practices, but none exists for TB IPC. One participant mentioned that

*As there was no implementation of TB IPC in the hospital, no monitoring and feedback systems were observed.*

**Workload, staffing, and bed occupancy.** The study participants mentioned that they had insufficient human resources and, therefore, they were overburdened and had to serve a high number of patients every day. The document review showed that the hospitals had a bed occupancy rate of more than 150 annually (Table 1). The number of patients reportedly always exceeded the number of beds in inpatient wards and occasionally increases up to 10 folds on admission days. One of the key informants mentioned that he had been using the space of 12 patients for 65 patients during the study period. Due to the excessive number of patients in a limited space, it was reportedly impossible to maintain a minimum spacing of one meter between patients.

*Due to the high patient load, we cannot follow standard IPC while dealing with pulmonary TB patients. Pulmonary TB patients stay with patients without TB in the inpatient wards. In most cases, we refer TB patients to TB specialty hospitals. However, the TB specialized hospitals often send them back to our hospital, or the patients themselves come back to our hospital and make the ward crowd.*

**Built environment, materials, and equipment for TB IPC.** All the participants mentioned that the inpatient wards had been spacious with large doors and windows; however, the renovation of the old buildings, the establishment of new buildings, and high patient load affected the open space, inpatient wards layout, cross ventilation, and access to sunlight. The participants added that they could not follow the standard IPC practice while dealing with infectious patients due to limited space and supply of IPC resources. One of the participants who was in a senior position at the hospital mentioned-

*We need facilities and resources for TB IPC implementation. We would require a separate space to receive presumptive TB patients, separate room to isolate them so that other general patients cannot be exposed to the presumptive TB patients.*

Both the hospitals were naturally ventilated. The nurses informed us that they often had to close doors and windows to avoid visitors and family caregivers in the ward. Moreover, they often kept the doors and windows closed to allow patients on the floor near the doors and windows. To avoid direct sunlight, some patients often preferred covering the windows with a curtain that hindered airflow. In the evening, the nurses reportedly closed the windows to prevent mosquitoes. They also shared that the patients or the ancillary workers often closed the doors and windows during the winter months to avoid cold winter air.

The study participants informed us that there was no supply of masks for patients in the hospital. HCW often recommended the patient with pulmonary TB should buy a mask. The participants informed us that while there was a supply of medical masks for HCW, staff had to raise a requisition for masks to the hospital store. There was no supply of N95 respirators.

The key informants also informed us that the hospitals lacked an electronic medical record (EMR) system, which delays TB/patient management. The participants mentioned that the documentation of patients' information is done manually using pen and paper, leading to a

long queue of patients in the outpatient department and diagnostic facilities. One of the key informants described the scenario as-

*Every morning, there was a long queue of patients to buy consultation tickets near the outpatient department. Once the patient visits the doctor and receives a prescription for any pathological or X-ray tests, the patient needs to wait in another long queue in the diagnostic facilities. Due to the long waiting time in multiple queues, some patients cannot complete all the tests on the same day. If we could digitalize the system, we could minimize patients hang around time and contamination of hospital areas with infectious droplets and airborne bacteria.*

## Healthcare workers-level factors

**Perception of nosocomial TB.** All the participants from KIIs and FGDs said they were at increased risk of TB infection due to prolonged and continued exposures in inpatient wards. The participants argued that HCW visits TB patients almost every day for 365 days a year. Most of the participants provided examples of TB disease among HCW, including multi-drug resistant TB. One of the key informants from a medicine ward mentioned that nurses were at the highest risk of TB infection due to their direct and long exposures to TB patients. Similar findings were also reflected in an FGD with nurses. One of the FGD participants said,

*We are at 100% risk of TB infection. There is no TB screening for presumptive TB patients in this hospital. There are always 1 to 4 TB patients in the ward. We do not take precautions, such as wearing masks when dealing with presumptive TB patients. Therefore, we are always at risk of TB infection.*

The HCW from the paediatric wards mentioned that they were at lower risk of nosocomial TB. They argued that paediatric patients could not produce cough and therefore were less likely to contaminate the air when compared with adult pulmonary patients. Consequently, they were not concerned about TB infection and did not take any precautions to prevent TB transmission. They reportedly visited presumptive TB patients without masks or respirators.

**Social and cultural aspects of TB and stigma.** TB was still considered a stigmatized disease. The participants mentioned that HCW with TB disease faced discrimination in the family, society, and workspace. To avoid stigma and discrimination, HCW often hides their TB disease. One of the key informants (nursing-in-charge) highlighted how and why HCW with TB often hide their TB status-

*Doctors, nurses, and allied HCW are getting infected with TB. They often hide that they have TB. To my knowledge, there are five nurses with TB disease in the hospital. They requested me not to tell their TB disease to others. They are unmarried girls. People do not want to marry girls with TB disease; even they become disease-free after treatment. Their family members also avoid them. Their colleagues also avoid them in the workspace. To avoid discrimination and stigma, HCW often hides their TB disease.*

## The reported role that healthcare workers can play in implementing TB IPC

The participants pointed out several factors that can be followed to better implement the TB IPC guidelines (Table 2). The nurses said that the doctors take a patient history and may be

**Table 2. Participants' recommendations for TB IPC implementation in health settings.**

| TB IPC measures | Recommendations |
|---|---|
| **Administrative** | |
| Guideline development | • Engage frontline caregivers in national TB IPC guideline development. |
| TB IPC committee | • Form a committee involving HCW from different professions and levels to facilitate and accelerate IPC activities in the hospital. |
| | • The hospital director, nursing superintendent, and ward-in-charge could be added to the hospital TB IPC committee. |
| | • Infection control coordination body |
| Cough screening | • Set up screening at the outdoors and emergency departments. |
| | • Before admitting a TB patient to the inpatient ward, there should be a screening system. |
| Separation | • In the inpatient ward, allocate two beds for presumptive TB patients where sufficient cross ventilation can be ensured. |
| | • Open a unit or ward for TB patients in tertiary care hospitals. |
| | • Establish a waiting area near the outdoors. |
| Isolation | • Presumptive TB patients can be sent to an isolation ward based on symptom screening. |
| Crowd control | • Control visitors |
| | • Allow patients according to bed numbers. |
| | • Initiate identity cards for visitors. |
| | • Only one adult visitor per patient can be allowed. |
| Training | • Training for trainers |
| | • Hospital-based training. |
| | • Ensure participation from all wards. |
| | • Annual refresher training |
| | • Using audio-visuals, educate patients and family caregivers about transmission pathways of TB, cough hygiene. |
| | • A monthly discussion on IPC among HCW |
| Infrastructure | • TB specialty hospitals should be enriched so that they can treat TB patients with comorbidities. |
| **Environmental** | |
| Ventilation | • Ensure opening the doors and windows. |
| | • Improve inpatient ward layout to ensure proper ventilation. |
| **Respiratory protection** | • Mask can be provided to presumptive TB patients on admission. |

aware of presumptive TB patients. The doctors may inform the nurses about presumptive TB patients, and the nurses can take necessary TB IPC measures. The nurses also mentioned that they could put a sign near a TB patient bed so that HCW can take standard precautions while visiting a TB patient. The nurses also added that they could use curtains around presumptive TB patients' beds as they have space limitations for isolation. One of the key-informant said-

*The patients enter the hospital through outdoors and emergency departments. We want TB IPC implementation, including cough screening at these two places. We also need to introduce the TB IPC control measures among the senior and junior level HCW. There should be frequent training so that HCW can attain the training by rotation. We also need infrastructure for TB screening and isolation. There is a need to collaborate with the government, hospital, and private sectors to reduce TB infection in the hospital.*

Lastly, it was suggested that local hospital HCW should be invited to the national TB IPC guidelines development committee. He said,

*Because HCW work at the local level and implement guidelines, their opinions should be reflected in guidelines. The central policymakers develop guidelines based on their perceptions and theoretical knowledge. Due to the lack of local level HCW views in the policy, the hospital contexts may not be reflected in the policy. Therefore, the policy may fail during the implementation stage.*

## Discussion

This study identified gaps in the TB IPC policies, hospital infrastructure, IPC supplies and equipment, which impact the TB IPC implementation. The findings showed that due to low levels of awareness about the national TB IPC guidelines, both the hospitals were not adhering to TB IPC and surveillance program, which is consistent with a nationally representative survey that found facilities lacked 85% of standard precaution components required for TB management and IPC in Bangladesh [20]. In the study facilities, identifying presumptive TB patients on admission was hindered by a lack of cough screening and a case definition. Separation and isolation of presumptive TB patients were affected by limited space, high patient load, limited supply of consumable IPC resources such as masks and respirators, HCW' IPC training gaps, and absence of isolation guidelines.

This study identified gaps in national TB IPC guideline recommendations and the hospital context. The guideline recommendations were based on tenuous theoretical principles informed by evidence from high-income low TB burden countries and underestimated resources required for implementation. Due to a lack of dedicated TB wards and clinics, TB patients were in all the adult medicine wards in the study hospitals. As outlined in the WHO's guidelines on Core Components of IPC, national IPC programs must ensure that infrastructure and IPC resources are available to implement IPC measures successfully [16]. The challenge is translating these recommendations to hospitals that have high bed-occupancy rate of more than 150%, overcrowding on admission days, and understaffing. Since these hospitals receive funding according to the bed capacity, it is likely that they continue to exhaust their resources and so cannot meet the TB IPC needs of staff when caring for the extra patients.

To assist TB IPC interventions and to detect TB outbreaks among patients and HCWs, WHO recommends healthcare associated TB infection surveillance in health settings [7]. While both study hospitals have laboratory capacities to support healthcare associated TB surveillance, it is not occurring. While this was beyond the scope of the project to explore the perspectives of those in government, it may be speculated that awareness about the burden of nosocomial TB in tertiary care hospitals may be low, leading to a failure to prioritize the issue. If surveillance data was available, it could be shared with NTP, hospital and department managers to motivate them to allocate resources for TB-specific IPC programs.

For TB care and control, introducing electronic medical records is one of the WHO's priority that may provide important backing to different components of WHO's End TB strategy's pillars 1 and pillars 2 [21–23]. Paper based recording and reporting increase diagnostic delay, which ultimately increases the length of hospital stay for pulmonary TB patients [24]. From our previous work, we identified that it took around five days to diagnose a patient with TB from admission, while the median length of hospital stay for a pulmonary TB patient without TB treatment was 5.4 days [3]. In the two study hospitals, we identified several key factors that are leading to the delay in diagnosing TB including: (1) a high number of patients; (2) patient

admission after office hours; (3) opening time of diagnostic facilities (8:30 AM to 3 PM), and (4) the lack of an EMR system. In alignment with previous calls for action, we feel that it is critical that action be taken to reduce the time taken to diagnose untreated pulmonary TB patients [24]. As recommended by the participants, an EMR system could help to minimize the time these patients spend at the patient registration desk and at diagnostic facilities, as well as helping to improve reporting times. The implementation of EMR systems is aligned with the Bangladesh government's national policy [25].

The WHO Core Components of IPC guidelines recommends that HCWs should receive education and training on IPC guidelines recommendations [7]. Amongst our participants, none reportedly received any education and training on TB IPC. A few HCWs reportedly received training on clinical management of TB patients that included TB IPC. However, the discussion on TB IPC in training might not be sufficient to motivate the HCWs to use their training to implement TB IPC in the hospitals. Due to limited training dedicated to TB IPC along with a lack of hospital-level TB IPC guidelines, there was no or minimal planning or execution of the IPC measures in the study hospitals The lack of training among nurses and ancillary workers was perceived as barriers to TB IPC; several participants called for the TB IPC guidelines and training on recommendations. Organizing training at these hospitals, promoting core and TB IPC and engaging all HCW, including doctors, nurses, and ancillary workers, may build partnership in the IPC program to help to implement the TB IPC measures.

In resourced limited settings, the WHO guidelines on TB IPC recommends natural ventilation in health settings [7]. Our findings showed that natural ventilation in the hospitals was affected by renovation and redistribution of internal space, cold winter air, mosquito nuisance, and by patients on the floor near doors and windows. To prevent airborne infection in the inpatients wards, doors and windows should be kept open day and nights and in all seasons [26]. On some occasions, natural ventilation can be more effective than mechanical ventilation in maintaining airflow and reducing TB transmission [7, 27]. We recommend that medicine, gynaecology, and obstetrics wards keep doors, windows, and other openings open until midnight in all seasons [28]. To ensure adequate ventilation, ancillary workers can be engaged to open the doors and windows in the morning, and nurses can be engaged for monitoring [27, 29]. Moreover, stickers/signage depicting 'keep doors/windows open' on the doors and windows and health education on the benefit of keeping doors and windows open can be given to patients and their family caregivers. Family caregivers can also be potential partners to implement natural ventilation intervention [30]. Large fans can be used above the doors and windows to maintain cross-ventilation to ensure the minimum air change per hour. Moreover, airbricks, window shutters etc., can stop the rain from coming in and control the cold breeze during the winter season [5]. Hospital stakeholders should address the factors that hindered opening doors and windows to maintain a minimum air change of six to 12 per hour recommended by WHO as airborne precaution [31].

Risk perceptions towards TB disease can also have an impact on acceptance and use of TB IPC measures [32]. Whilst our participants were aware that they are being exposed to pulmonary TB patients and that they were at risk of TB infection; their risk perception did not translate into action. Limited understanding about disease transmission and poor risk perception may result in non-compliance, poor health-seeking behavior, and treatment that may result in TB transmission in the hospital [33]. The perception amongst participants that health workers could be a source of TB transmission in the hospital was poor, with many HCW continuing to work when they were sick or hide their TB disease status while working in the hospital. The reasons for continuing working while they were still sick could be their fear of losing salary when they will be on sick leave; and fear of being re-assigned or even fired as noted by Delft et al [34]. The reason to hide TB status could be due to the fear of being stigmatized by co-

workers. In a study among 879 HCW in South Africa, 19% of the respondents agreed that HCWs with presumptive TB are stigmatized in hospitals [35]. The HCW with TB disease can be a source of infection in the inpatient ward and can affect the effectiveness of TB IPC measures if implemented. The WHO 2019 TB IPC guidelines recommend educating HCWs, patients, family caregivers and community members about causes of stigma and discrimination that can be followed in public tertiary care hospitals to reduce TB-related stigma [7].

This study has limitations. We conducted this study in only two hospitals, and therefore, the study findings are very context specific. The findings presented study can be an indicative of hospital preparedness for TB IPC implementation in similar hospitals in Bangladesh, and other low-income high TB burden countries. Moreover, the study findings are consistent with a prior study that assessed the TB IPC in 11 TB specialized hospitals, tertiary care hospitals, and TB clinics [2]. Secondly, we did not interview any stakeholders from the ministry of health or NTP. Therefore, we could not understand the national-level factors that may influence TB IPC implementation in the local level tertiary care hospitals. Finally, this assessment identified strengths and weaknesses in health systems and services to adopt TB IPC, and the context may change over time and impact effectiveness.

In conclusion, this study found that while there was a positive attitude towards TB IPC and staff members are willing to implement strategies, there are key barriers that need to be addressed. Currently the study hospitals will struggle to implement the TB IPC measure due to high patient load, limited space, and lack of TB IPC specific budget and resources. Establishing new hospitals to reduce patient load is time and resource expensive. Therefore, we recommend leveraging existing resources for the implementation of TB IPC measures. We recommend that the space limitation for presumptive TB patient isolation be minimized by setting up an isolation area in the veranda or corridors with a large open area facing outside. If it is not possible during the rainy monsoon season, a curtain can separate patients inside the inpatient ward. Prior studies showed that the use of curtains had ben efficient in reducing the spread of TB bacteria during coughing, sneezing, and other aerosol-generating activities [36]. All coughing patients should wear a mask throughout while in hospital or until TB disease is ruled out as well other contagious respiratory infections. Besides, the ancillary workers can be engaged in IPC programs to keep the doors and windows open to ensure natural ventilation. The national TB program or other national and international organizations working on TB elimination can visit these hospitals, form TB IPC committees and organize TB IPC training in the hospitals so that the HCW fully understand TB IPC guidelines recommendations and build the task into their daily activities. Some nurses or senior nursing students can be trained on TB advocacy and counselling to provide health education, including respiratory hygiene and counselling, to support TB patients and their family caregivers. All HCW should be given information and motivation, including a care package, so they feel encouraged to undergo TB diagnostic investigation if they have signs and symptoms suggestive of TB. Finally, we recommend public tertiary care hospitals receive a budget according to the annual number of patients, not according to the allocated beds for the hospitals in Bangladesh.

## Supporting information

**S1 Appendix. Guidelines for key informant interviews.**
(DOCX)

**S2 Appendix. Focus group discussion guideline.**
(DOCX)

**S1 Data.**
(DOCX)

# Acknowledgments

We want to thank the study hospitals' directors and all the study participants for their time and respect. icddr,b acknowledges with gratitude the commitment of CDC to its research efforts. icddr,b is also grateful to the Governments of Bangladesh, Canada, Sweden, and the UK for providing core/unrestricted support.

# Author Contributions

**Conceptualization:** Md. Saiful Islam, Sayera Banu, Abrar Ahmad Chughtai, Holly Seale.

**Data curation:** Md. Saiful Islam, Sayeeda Tarannum, Kamal Ibne Amin Chowdhury.

**Formal analysis:** Md. Saiful Islam, Sayeeda Tarannum, Holly Seale.

**Funding acquisition:** Md. Saiful Islam, Sayera Banu.

**Investigation:** Md. Saiful Islam.

**Methodology:** Md. Saiful Islam, Kamal Ibne Amin Chowdhury, Abrar Ahmad Chughtai, Holly Seale.

**Project administration:** Md. Saiful Islam, Sayeeda Tarannum, Sayera Banu, Kamal Ibne Amin Chowdhury, Arifa Nazneen.

**Supervision:** Sayera Banu, Arifa Nazneen, Abrar Ahmad Chughtai, Holly Seale.

**Validation:** Md. Saiful Islam, Sayeeda Tarannum, Sayera Banu, Kamal Ibne Amin Chowdhury, Abrar Ahmad Chughtai.

**Visualization:** Holly Seale.

**Writing – original draft:** Md. Saiful Islam.

**Writing – review & editing:** Md. Saiful Islam, Sayeeda Tarannum, Sayera Banu, Kamal Ibne Amin Chowdhury, Arifa Nazneen, Abrar Ahmad Chughtai, Holly Seale.

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
