## [Decision Letter · Decision Letter 0]

1 Dec 2021

PONE-D-21-24921Exploring hospital readiness for implementing the National TB infection control guidelines in two public tertiary care hospitals, BangladeshPLOS ONE

Dear Dr. Islam,

Thank you for submitting your manuscript to PLOS ONE. After careful consideration, we feel that it has merit but does not fully meet PLOS ONE’s publication criteria as it currently stands. Therefore, we invite you to submit a revised version of the manuscript that addresses the points raised during the review process.

We look forward to receiving your revised manuscript.

Kind regards,

Rabia Hussain

Academic Editor

PLOS ONE

Journal Requirements:

3. Please include your tables as part of your main manuscript and remove the individual files. Please note that supplementary tables (should remain/ be uploaded) as separate "supporting information" files"

4. Thank you for stating the following in the Acknowledgments / Financial Support Section of your manuscript: 

This research protocol was funded by the United States Centre for Disease Control and Prevention (CDC) through the cooperative agreement grant number 5U01GH1207. The funders had no role in study design, data collection and analysis, decision to publish, or manuscript preparation.

Please remove any funding-related text from the manuscript and let us know how you would like to update your Funding Statement. Currently, your Funding 

This research protocol was funded by the United States Centre for Disease Control and Prevention (CDC) through the cooperative agreement grant number 5U01GH1207. The funders had no role in study design, data collection and analysis, decision to publish, or manuscript preparation.

6. We note that you have referenced (ie. Bewick et al. [5]) which has currently not yet been accepted for publication. Please remove this from your References and amend this to state in the body of your manuscript: (ie “Bewick et al. [Unpublished]”) as detailed online in our guide for authors

Reviewers' comments:

Reviewer's Responses to Questions

**Comments to the Author**

1. Is the manuscript technically sound, and do the data support the conclusions?

Reviewer #1: Yes

Reviewer #2: Partly

Reviewer #3: Yes

2. Has the statistical analysis been performed appropriately and rigorously? 

Reviewer #1: Yes

Reviewer #2: Yes

Reviewer #3: N/A

3. Have the authors made all data underlying the findings in their manuscript fully available?

Reviewer #1: No

Reviewer #2: Yes

Reviewer #3: Yes

4. Is the manuscript presented in an intelligible fashion and written in standard English?

Reviewer #1: Yes

Reviewer #2: Yes

Reviewer #3: Yes

5. Review Comments to the Author

Reviewer #1: General comments

This manuscript has addressed an important aspect about Tuberculosis Infections Prevention and Control at hospital settings in a high TB burden country-Bangladesh. The authors used multiple methodological approach (Triangulation) to collect rich data/information hence presented a reliable findings. The authors used a rigorous analysis and results and discussion sections were logically narrated.

On the other hand, the author did not insert lines in the manuscript for easy review. Please amend this in the next review.

Specific comments

Title:

The title lacks methodological design and the word "hospital" is repeated. I suggest the title reads:

Full title: Readiness of tertiary care hospitals to implement the National TB infection prevention and control guidelines in Bangladesh: A qualitative exploration

Short title: A qualitative exploration of readiness of hospitals to implement Tuberculosis guidelines

Abstract:

- The abstract contains many but necessary abbreviations! If it is possible the authors should minimize the number of abbreviations.

Others:

- The authors should add a paragraph to explain how is TB specialized hospital differs from tertiary hospital. Does TB specialized hospital implement the National TB IPC guideline?

- On page 16: A sentence "This situation has probably been exacerbated due to the COVID-19 pandemic, with resources being diverted towards pandemic management". This study was conducted from January to December 2019 before the COVID-19 pandemic. Therefore, the author should consider omitting the sentence.

- On page 19: a sentence - All coughing patients should wear a mask "until they confirm the diagnosis". Since the aim is to prevent TB infection, I suggest the author to rephrase the later part of the sentence. It should read "throughout while in hospital or until TB disease is ruled out as well other contagious respiratory infections".

Reviewer #2: Since the National Tuberculosis Policy is there in country since long , readiness is not a suitable word , instead authors could have used the phrase bottlenecks in implementation of Latest NTP IPC and highlight the lacunas as per the guidelines . In the abstract"However, there has been a failure to date to assess" needs to be rephrased . Also in main manuscript there are sections which needs better rephrasing. In the reference section ref number 1 and 7 are not published , until they are visible online how can they be cited . Other details have been incorporated in the pdf attached under comments .

Reviewer #3: Islam et al have used a qualitative study design to explore hospital readiness for national TB infection control in two public hospitals in Bangladesh. They used a review of national TB infection control policies, key-informant interviews and four focus group discussions with HCWs to understand knowledge,attitude, and practice of TB IPC measures. They found a lack of preparedness, resources, training, and knowledge about TB IPPC practices including of national IPC guidelines in both hospitals. They highlight several factors that limit the ability to implement an infection control program, including overcrowding in wards, built environment and limited IPC resources. They provide suggestions on what steps need to be taken to facilitate implementation of IPC programs.

General comments: This paper explores an underappreciated aspect of TB control and highlight important issues that need to be addressed. The methodology is reliable to answer the questions being asked in this study, although the selection participants selected for further interviews appears to be quite arbitrary, especially with the use of seniority as a proxy for expertise. There is a blurring between issues that are unique to TB infection control and those that affect clinical care in general (resource shortage and electronic medical records). The paper would benefit from highlighting gaps and overalps between concerns identified by the study participants and priorities identified by the WHO. It would be helpful to discuss issues of TB IPC unique to outpatient clinics and those unique to the inpatient wards. The discussion needs to be shorter and focused on highlighting the findings from their study and contextualizing them with TB control in Bangladesh. I would strongly recommend against making policy proposals in the discussion as that was not the focus of the study, and because those policies already exist.

Specific comments:

Introduction

Reference 3 appears to be a self-citation of a study done in the same 2 hospitals. Are there additional studies highlighting rates of latent TB infection in other hospitals in Bangladesh? Could those be included as well?

Materials and methods:

How many participants were screened for interview before arriving at the 51 who were included in the study?

Results

There is a mention of frequent training that some doctors received - how is that those doctors were not aware of the national TB IPC policy, and why were the doctors unable to use their training to implement infection control activities?

Do the public hospitals have dedicated TB wards and clinics? Is there a reason that TB patients are located in all the wards.

The limitations on ventilation imposed by overcrowding as well as by hot and cold weather are common in public hospitals across LMIC nations worldwide and raise important issues that need to be addressed. This should be highlighted in the discussion as this is often underestimated by policy makers.

Discussion:

The discussion needs significant rewriting. The authors should outline their findings which are perceptions and opinions of the participants. It appears that the authors have used those findings to make suggestions about changes that need to be implemented (financing, laboratory services, delays in diagnosis, need for EMR systems) that are not unique to TB. Using the participants opinions to frame policy is unwise, esepecially since the participants have clearly stated they are unaware of the national TB IPC program that likely addresses these issues. Some of the recommendations are quite generic ( improving communication) and speculative. It would be best not to use the discussion to make policy recommendations.

The focus of the discussion should be on TB IPC policy in Bangladesh. Can the authors use their findings to identify priorities and a roadmap for implementation of a TB IPC program in public hospitals in Bangladesh? Are their unique aspect that have been identified that are not mentioned in the national policy? Could the authors discuss IPC practices in TB specialty hospitals in Bangladesh and how that compares with public hospitals?

There is a mention of frequent training that some doctors received - how is that those doctors were not aware of the national TB IPC policy, and why were the doctors unable to use their training to implement infection control activities? Is the issue a lack of planning or a lack of execution?

The authors highlight the cognitive dissonance common in LMICs where HCWs do not wear masks despite being exposed to TB and having high rates of latent TB and active TB. They also have to deal with the stigma that forces them to hide their diagnosis from their colleagues, including working while possibly infectious and contributing to the spread of TB.

6. PLOS authors have the option to publish the peer review history of their article (what does this mean?). If published, this will include your full peer review and any attached files.

Reviewer #1: **Yes: **Festo K. Shayo

Reviewer #2: **Yes: **Rishabh Kumar Rana

Reviewer #3: No

---

## [Author Response · Author response to Decision Letter 0]

9 Dec 2021

PONE-D-21-24921

Exploring hospital readiness for implementing the National TB infection control guidelines in two public tertiary care hospitals, Bangladesh

PLOS ONE

Date: December 12, 2021

To

Rabia Hussain

Academic Editor

PLOS ONE

Re - Exploring hospital readiness for implementing the National TB infection control guidelines in two public tertiary care hospitals, Bangladesh- PONE-D-21-24921

Dear Rabia Hussain:

We are thankful to the reviewers for their valuable feedback. We are also grateful to the editor for allowing us to respond to the comments. Based on the helpful feedback, we revised the manuscript and believe it is more precise, clear, and informative. The following is an itemized list of our specific responses to the reviewers’ comments. We have highlighted where the changes have been made in the marked version of the manuscript. 

We would appreciate your further review. Please contact me directly with any additional questions or comments. We look forward to hearing from you.

Sincerely

Md. Saiful Islam

Corresponding Author

saiful@icddrb.org

Comment: Journal Requirements:

Response: Thank you. The manuscript and the file names have been updated according to PLOS ONE’s style requirements.

Comment: 2. Please include additional information regarding the survey or questionnaire used in the study and ensure that you have provided sufficient details that others could replicate the analyses. For instance, if you developed a questionnaire as part of this study and it is not under a copyright more restrictive than CC-BY, please include a copy, in both the original language and English, as Supporting Information.

Response: Thank you. We used standard guidelines for interview and focus groups discussion. We discussed these as “Through KII, the team explored (i) hospital infrastructure, context, and readiness, (ii) HCW risk perception, concerns, attitudes and anticipated barriers regarding the TB IPC implementation; and (ii) the role that HCW (and others) can play in overcoming the barriers” on page six lines 125-127.

On page 6 lines 135-137, we added, “We convened the FGDs to elicit HCW perspectives on more complex phenomena (e.g., power relation, hospital priority, social and cultural aspects of TB, and TB among healthcare workers)”.

As recommended, we uploaded data collection tools as supporting information.

Comment: 3. Please include your tables as part of your main manuscript and remove the individual files. Please note that supplementary tables (should remain/ be uploaded) as separate "supporting information" files"

Response: Thank you. The tables have been included as part of the main manuscript. 

Comment: 4. Thank you for stating the following in the Acknowledgments / Financial Support Section of your manuscript: 

This research protocol was funded by the United States Centre for Disease Control and Prevention (CDC) through the cooperative agreement grant number 5U01GH1207. The funders had no role in study design, data collection and analysis, decision to publish, or manuscript preparation.

Please remove any funding-related text from the manuscript and let us know how you would like to update your Funding Statement. Currently, your Funding 

This research protocol was funded by the United States Centre for Disease Control and Prevention (CDC) through the cooperative agreement grant number 5U01GH1207. The funders had no role in study design, data collection and analysis, decision to publish, or manuscript preparation.

Response: Thank you. We have checked the acknowledgments sections and removed any funding information from the acknowledgments section.

Comment: 5. In your Data Availability statement, you have not specified where the minimal data set underlying the results described in your manuscript can be found. PLOS defines a study's minimal data set as the underlying data used to reach the conclusions drawn in the manuscript and any additional data required to replicate the reported study findings in their entirety. All PLOS journals require that the minimal data set be made fully available. For more information about our data policy, please see http://journals.plos.org/plosone/s/data-availability.

Response: As recommended, we shared a minimal dataset as supplementary information.

Comment: 6. We note that you have referenced (ie. Bewick et al. [5]) which has currently not yet been accepted for publication. Please remove this from your References and amend this to state in the body of your manuscript: (ie “Bewick et al. [Unpublished]”) as detailed online in our guide for authors

Response: Thank you. We have removed all the unpublished manuscripts as a reference from the manuscript.

Reviewers' comments:

Reviewer #1: 

General comments: This manuscript has addressed an important aspect about Tuberculosis Infections Prevention and Control at hospital settings in a high TB burden country-Bangladesh. The authors used multiple methodological approach (Triangulation) to collect rich data/information hence presented a reliable finding. The authors used a rigorous analysis and results, and discussion sections were logically narrated.

Response: Thank you so much for your appreciation. 

Comment: On the other hand, the author did not insert lines in the manuscript for easy review. Please amend this in the next review.

Response: We are sorry for this mistake. We have now added line numbers in this revised version.

Comment: Specific comments

Title: The title lacks methodological design and the word "hospital" is repeated. I suggest the title reads: Full title: Readiness of tertiary care hospitals to implement the National TB infection prevention and control guidelines in Bangladesh: A qualitative exploration

Short title: A qualitative exploration of readiness of hospitals to implement Tuberculosis guidelines

Response: Thank you. Based on your and other reviewers’ comments, we have revised the full and short titles. Now it reads-

Full title: Preparedness of tertiary care hospitals to implement the National TB infection prevention and control guidelines in Bangladesh: A qualitative exploration

Short title: A qualitative exploration of preparedness of hospitals to implement Tuberculosis guidelines

Comment: Abstract: - The abstract contains many but necessary abbreviations! If it is possible the authors should minimize the number of abbreviations.

Response: Based on your recommendation, we have revised the abstract and deleted a few abbreviations.

Comment: Others:- The authors should add a paragraph to explain how is TB specialized hospital differs from tertiary hospital. Does TB specialized hospital implement the National TB IPC guideline?

Response: Thank you. We now revised the introductory paragraph and added, “In Bangladesh, TB specialized hospitals admit and treat patients with drug-susceptible and drug-resistant TB as inpatients. Prior studies showed that TB specialized hospitals implement TB IPC guidelines on a regular basis[1]. On the other hand, tertiary care general hospitals admit presumptive TB patients for diagnosis and occasionally provide treatment to TB patients with comorbidities. Prior studies document that pulmonary TB patients stay on average 5.5 days in public tertiary care general hospitals and there was very limited implementation of TB IPC measures in the hospitals” on page 3 lines 53-57.

Comment: On page 16: A sentence "This situation has probably been exacerbated due to the COVID-19 pandemic, with resources being diverted towards pandemic management". This study was conducted from January to December 2019 before the COVID-19 pandemic. Therefore, the author should consider omitting the sentence.

Response: As suggested, we have deleted this sentence, “This situation has probably been exacerbated due to the COVID-19 pandemic, with resources being diverted towards pandemic management [2, 3]” on page 21, lines 385-387.

Comment- On page 19: a sentence - All coughing patients should wear a mask "until they confirm the diagnosis". Since the aim is to prevent TB infection, I suggest the author to rephrase the later part of the sentence. It should read "throughout while in hospital or until TB disease is ruled out as well other contagious respiratory infections".

Response: Thank you. We have revised the sentence: Now it reads, “All coughing patients should wear a mask while in the hospital or until TB disease is ruled out as well other contagious respiratory infections” on page 25, lines 490-492.

Reviewer #2: 

Comment: Since the National Tuberculosis Policy is there in country since long , readiness is not a suitable word , instead authors could have used the phrase bottlenecks in implementation of Latest NTP IPC and highlight the lacunas as per the guidelines . 

Response: Thank you. Based on your and other reviewers’ comments, we have revised the full and short titles. Now it reads-

Full title: Preparedness of tertiary care hospitals to implement the National TB infection prevention and control guidelines in Bangladesh: A qualitative exploration

Short title: A qualitative exploration of preparedness of hospitals to implement Tuberculosis guidelines

Comment: In the abstract "However, there has been a failure to date to assess" needs to be rephrased. 

Response: Based on your comments, we have revised the sentence. Now it reads, “However, there has been no published literature to date that assessed the preparedness of hospitals to comply with the recommendations.” On page 2, line 30.

Comment: Also in main manuscript there are sections which needs better rephrasing.

Response: Based on your comments on the manuscript, we revised the manuscript. We have replaced the word “readiness” with “preparedness.” Please see below our responses to the specific comments provided in the manuscript. 

Comment: In the reference section ref number 1 and 7 are not published, until they are visible online how can they be cited. Other details have been incorporated in the pdf attached under comments.

Response: Reference numbers 1 and 7 have now been published or in press. We have updated the reference.

Comment: Authors in their introduction note that Bangladesh has developed its guidelines in 2011. So, when they say that readiness has not been assessed, it is not suitable as the program is already there, they should rephrase it rather to something like “It has not been assessed regarding strict adherence.”

Response: Based on your suggestion, we have revised the sentence. Now it reads, “However, there has been no published literature to date that assessed the preparedness of hospitals to comply with the recommendations” on page 4, lines 77-79.

Comment: Gaps were identified in the Hospital; TB IPC Policy faults or Gaps were not mentioned in either discussion or methods. As is clear most of the staff were unaware about the existence of IPC guidelines for TB.

Response: Thank you. Under methods, we added, “ TB“TB IPC policies and historical documents were reviewed to understand gaps in the policy contents, hospital characteristics, building design, in-patient-ward layout, and renovation, …” on page 8, lines 144-145.

Under results, we added, “The policy review found gaps in the content of the guidelines and many of the guideline recommendations were not context-appropriate. For example, the national TB IPC guidelines recommend triage and isolation of presumptive TB patients. However, none of the hospitals had dedicated TB wards or isolation rooms where presumptive TB patients could be isolated.” on pages 10-11, lines 191-195.

Under discussion, we added, “This study identified gaps in national TB IPC guideline recommendations and the hospital context. The recommendations were based on tenuous theoretical principles informed by evidence from high-income low TB burden countries and underestimated resources that are required for implementation. Due to a lack of dedicated TB wards and clinics, TB patients were in all the adult medicine wards.” on page 20, lines 375-379.

Comments: It should be rephrased to hospitals were not adhering to IPC guidelines primarily due to unawareness.

Response: Thank you. We have revised the sentence. Now it reads, “The findings showed that due low levels of awareness about the national TB IPC guidelines, both the hospitals were not adhering to TB IPC and surveillance program,…..on page 20 lines 365-367”.

Reviewer #3: 

Comment: Islam et al have used a qualitative study design to explore hospital readiness for national TB infection control in two public hospitals in Bangladesh. They used a review of national TB infection control policies, key-informant interviews and four focus group discussions with HCWs to understand knowledge, attitude, and practice of TB IPC measures. They found a lack of preparedness, resources, training, and knowledge about TB IPPC practices including of national IPC guidelines in both hospitals. They highlight several factors that limit the ability to implement an infection control program, including overcrowding in wards, built environment and limited IPC resources. They provide suggestions on what steps need to be taken to facilitate implementation of IPC programs.

Response: Thank you so much for the constructive feedback.

Comment: General comments: This paper explores an underappreciated aspect of TB control and highlight important issues that need to be addressed. The methodology is reliable to answer the questions being asked in this study, although the selection participants selected for further interviews appears to be quite arbitrary, especially with the use of seniority as a proxy for expertise. 

Response: Thank you. We agree that the field team selected those with the highest duration working in tertiary care hospitals. However, we followed another step before confirming the participants for interview/discussion. On pages 5, lines 98-104, we mentioned, “To recruit participants for interviews and focus group discussions (FGD), the team made a list of HCW who were either members of hospital committees, or who were deemed to be influential in decision-making, knowledgeable, and experienced in dealing with infectious disease patients, or those staff who were seen as influence other’s practices”.

Comment: There is a blurring between issues that are unique to TB infection control and those that affect clinical care in general (resource shortage and electronic medical records).

Response: Thank you. Clinical management of pulmonary TB patients and TB IPC are interrelated. The introduction of electronic medical records could support improvements in clinical care in general. For TB care and control, introducing electronic medical records is one of the WHO’s priorities that may provide significant backing to different components of WHO’s End TB strategy’s pillars 1 and 2 [4-6]. We would like to stress that the resource shortage and lack of electronic records directly affect the TB IPC in these facilities. In a companion study[7], we found that lack of electronic medical records contributes to TB patients' length of hospital stay. Due to the lack of resources, HCWs could not provide face masks to patients or wear a respirator for themselves while treating presumptive TB patients.

Comment: The paper would benefit from highlighting gaps and overlaps between concerns identified by the study participants and priorities identified by the WHO. 

Response: Based on your suggestion, we have now updated the discussion section. 

On page pages 21, lines 388-392, we added, “ To assist TB IPC interventions and to detect TB outbreaks among patients and HCWs, WHO recommends healthcare-associated TB infection surveillance in health settings[8]. While both study hospitals have laboratory capacities to support healthcare-associated TB surveillance, it is not occurring….”

On page 21, lines 398-402, we added, “For TB care and control, introducing electronic medical records is one of the WHO’s priority and may provide important backing to different components of WHO’s End TB strategy’s pillars 1 and pillars 2[4-6]. Paper-based recording and reporting increase diagnostic delay, which ultimately increases the length of hospital stay for pulmonary TB patients [9]…..”

Comment: It would be helpful to discuss issues of TB IPC unique to outpatient clinics and those unique to the inpatient wards. 

Response: Thank you. We would like to inform you that we have not collected data separately for outpatient clinics and inpatient wards. It is beyond the scope of the current manuscript, and therefore, we could not separate the analysis.

Comment: The discussion needs to be shorter and focused on highlighting the findings from their study and contextualizing them with TB control in Bangladesh.

Response: Thank you. Based on your suggestion, we have tightened the discussion section and contextualized them with TB infection control in Bangladesh. 

Comment: I would strongly recommend against making policy proposals in the discussion as that was not the focus of the study, and because those policies already exist.

Response: We agree with you. We have modified the recommendations. We aimed to generate evidence and highlight the hospital context so that the policymakers consider these when revising or updating the national TB IPC policy.

Comments: Specific comments: Introduction

Reference 3 appears to be a self-citation of a study done in the same 2 hospitals. Are there additional studies highlighting rates of latent TB infection in other hospitals in Bangladesh? Could those be included as well?

Response: This is the only published study from Bangladesh discussing rates of latent TB infection among healthcare workers in public tertiary care general hospitals. 

Comment: Materials and methods: How many participants were screened for interview before arriving at the 51 who were included in the study?

Response: Thank you. We have now added the information. Now it reads, “The team conducted 16 KIIs and four FGDs. The field team approached 18 HCWs for KIIs and 38 HCWs for FGDs, where 16 HCW consented to participate as key informants, and 35 consented as FGD participants.” On page 10, lines 181-183.

Comments: Results. There is a mention of frequent training that some doctors received - how is that those doctors were not aware of the national TB IPC policy, and why were the doctors unable to use their training to implement infection control activities?

Response: On page 12, lines 216-218, we mentioned that none of the HCWs received dedicated TB IPC training. We also said, “Some doctors mentioned that they received education and training on TB patient treatment and management, and there were brief discussions on TB IPC.” Since the training was not focused on TB IPC, including the different TB IPC control measures, the doctors could not use their training to implement infection control activities. We have now discussed this under the discussion as “A few HCWs reportedly received training on clinical management of TB patients that included TB IPC. However, the discussion on TB IPC in training might not be sufficient to motivate the HCWs to use their training to implement TB IPC in the hospitals” on page 22, lines 415-420.

Comments: Do the public hospitals have dedicated TB wards and clinics? Is there a reason that TB patients are located in all the wards?

Response: None of the public hospitals had dedicated TB wards and clinics. Due to limited space and resources, TB patients are located in adult medicine wards. Under the result section, we added, “….., none of the hospitals had dedicated TB wards or isolation rooms where presumptive TB patients could be isolated” on page 11, lines 194-195. In the discussion, we added, “This study identified gaps in national TB IPC guideline recommendations and the hospital context. The guideline recommendations were based on tenuous theoretical principles informed by evidence from high-income low TB burden countries and underestimated resources required for implementation. Due to a lack of dedicated TB wards and clinics, TB patients were in all the adult medicine wards in the study hospitals” on page 20, lines 375-379.

Comment: The limitations on ventilation imposed by overcrowding as well as by hot and cold weather are common in public hospitals across LMIC nations worldwide and raise important issues that need to be addressed. This should be highlighted in the discussion as this is often underestimated by policymakers.

Response: Thank you. Based on your suggestion, we have extended our discussion on ventilation. We now added, “In resourced limited settings, the WHO guidelines on TB IPC recommends natural ventilation in health settings[8]. Our findings showed that natural ventilation in the hospitals was affected by renovation and redistribution of internal space, cold winter air, mosquito nuisance, and by patients on the floor near doors and windows. Doors and windows should be kept open day and night and in all seasons to prevent TB infection in the inpatients' wards[10]. On some occasions, natural ventilation can be more effective than mechanical ventilation in maintaining airflow and reducing TB transmission [8, 11]. We recommend that all adult medicine wards that host presumptive TB patients keep doors, windows, and other openings open until midnight in all seasons [12]. To ensure adequate ventilation, ancillary workers can be engaged to open the doors and windows in the morning, and nurses can be engaged for monitoring [11, 13]. Moreover, stickers/signage depicting ‘keep doors/windows open’ on the doors and windows and health education on the benefit of keeping doors and windows open can be given to patients and their family caregivers. Family caregivers can also be potential partners to implement natural ventilation intervention [14]. Large fans can be used above the doors and windows to maintain cross-ventilation to ensure the minimum air change per hour. Moreover, airbricks, window shutters, etc., can stop the rain from coming in and control the cold breeze during the winter season [15]. Hospital stakeholders should address the factors that hindered opening doors and windows to maintain a minimum air change of six to 12 per hour recommended by WHO as an airborne precaution[16]” on page 23 lines 431-449.

Comment: Discussion:

The discussion needs significant rewriting. 

Response: We have revised the discussion section extensively as suggested. 

Comment: The authors should outline their findings which are perceptions and opinions of the participants. 

Response: Thank you so much for this suggestion. We used multiple data collection tools and analyzed the data combinedly for thematic purposes. Separating the findings by perceptions and opinions would require re-analysis of the data, which is time and resource expensive. Therefore, we have decided not to re-analyze the data.

Comment: It appears that the authors have used those findings to make suggestions about changes that need to be implemented (financing, laboratory services, delays in diagnosis, need for EMR systems) that are not unique to TB. 

Response: While we agree with you that financing, laboratory services, delays in diagnosis, need for EMR systems are not unique to TB, however, these are affecting TB IPC implementation. The WHO also prioritizes these areas for TB IPC and core components of IPC measures[17].

Comment: Using the participants opinions to frame policy is unwise, especially since the participants have clearly stated they are unaware of the national TB IPC program that likely addresses these issues. 

Response: Thank you. We revised the recommendations.

Comment: Some of the recommendations are quite generic (improving communication) and speculative. It would be best not to use the discussion to make policy recommendations.

Response: Thank you. We have deleted this paragraph from the manuscript, “Beyond training, it is also important to support communication within the hospital. Our participants indicated that there was a communication gap between doctors, nurses, and other ancillary workers. This may be contributing to low uptake of IPC practices, particularly around the separation and isolation of presumptive TB patients. Communication between hospital leadership, doctors, nurses, and other ancillary workers can support adherence to IPC measures [18]. Prior studies have shown that receiving instructional feedback from senior colleagues can motivate other staff members to implement IPC measures [19]”on pages 22-23, lines 423-429.

For other recommendations, we provided references to support our recommendations.

Comment: The focus of the discussion should be on TB IPC policy in Bangladesh. Can the authors use their findings to identify priorities and a roadmap for the implementation of a TB IPC program in public hospitals in Bangladesh? 

Response: Thank you so much. We want to inform you that we are preparing another manuscript to discuss priorities and a roadmap for implementing a TB IPC program in public hospitals in Bangladesh. Therefore, we decided not to focus more on these under the current manuscript.

Comment: Are their unique aspect that have been identified that are not mentioned in the national policy? 

Response: Thank you. We identified several aspects that have not been mentioned in the national policy. For example, the national policy recommends the use of face masks for patients and N95 respirators for HCWs. However, there was no supply of these PPE for HCWs in public tertiary care general hospitals. The policy recommends triage and isolation of presumptive TB patients. All the public tertiary-care general hospitals lacked dedicated TB wards or clinics. There was no mention of how these hospitals should triage or isolate presumptive TB patients. We have discussed all these in the manuscript. These have also been discussed in the manuscript.

Comment: Could the authors discuss IPC practices in TB specialty hospitals in Bangladesh and how that compares with public hospitals?

Response: We appreciate your comments. As part of this study, we did not collect data from TB specialty hospitals. Therefore, it is beyond the scope of the current manuscript to compare results between TB specialty hospitals and general public hospitals.

Comment: There is a mention of frequent training that some doctors received - how is that those doctors were not aware of the national TB IPC policy, and why were the doctors unable to use their training to implement infection control activities? Is the issue a lack of planning or a lack of execution?

Response: On page 12, lines 216-218, we mentioned that none of the HCWs received dedicated TB IPC training. We also said, “Some doctors mentioned that they received education and training on TB patient treatment and management, and there were brief discussions on TB IPC.” Since the training was not focused on TB IPC, including the different TB IPC control measures, the doctors could not use their training to implement infection control activities. We have now discussed this under the discussion as “A few HCWs reportedly received training on clinical management of TB patients that included TB IPC. However, the discussion on TB IPC in training might not be sufficient to motivate the HCWs using their training to implement TB IPC in the hospitals. Due to limited training dedicated to TB IPC along with a lack of hospital-level TB IPC guidelines, there was no or minimal planning or execution of the IPC measures in the study hospitals ” on page 22, lines 415-420.

Comment: The authors highlight the cognitive dissonance common in LMICs where HCWs do not wear masks despite being exposed to TB and having high rates of latent TB and active TB. They also have to deal with the stigma that forces them to hide their diagnosis from their colleagues, including working while possibly infectious and contributing to the spread of TB.

Response: Thank you. We have discussed this, “The reason to hide TB status could be due to the fear of being stigmatized by co-workers. In a study among 879 HCW in South Africa, 19% of the respondents agreed that HCWs with presumptive TB are stigmatized in hospitals[20]. The HCW with TB disease can be a source of infection in the inpatient ward and can affect the effectiveness of TB IPC measures if implemented. The WHO 2019 TB IPC guidelines recommend educating HCWs, patients, family caregivers and community members about causes of stigma and discrimination that can be followed in public tertiary care hospitals to reduce TB-related stigma[8]” on page 21, lines 466-469.

References

1. Nazneen A, Tarannum S, Chowdhury KIA, Islam MT, Islam SMH, Ahmed S, et al. Implementation status of national tuberculosis infection control guidelines in Bangladeshi hospitals. PloS one. 2021;16(2):e0246923. doi: 10.1371/journal.pone.0246923.

2. Alene KA, Wangdi K, Clements ACA. Impact of the COVID-19 Pandemic on Tuberculosis Control: An Overview. Tropical medicine and infectious disease. 2020;5(3). Epub 2020/07/30. doi: 10.3390/tropicalmed5030123. PubMed PMID: 32722014; PubMed Central PMCID: PMCPMC7558533.

3. Burki T. Global shortage of personal protective equipment. The Lancet Infectious diseases. 2020;20(7):785-6. doi: 10.1016/S1473-3099(20)30501-6. PubMed PMID: 32592673.

4. Timimi H, Falzon D, Glaziou P, Sismanidis C, Floyd K. WHO guidance on electronic systems to manage data for tuberculosis care and control. J Am Med Inform Assoc. 2012;19(6):939-41. Epub 2012/06/22. doi: 10.1136/amiajnl-2011-000755. PubMed PMID: 22729390.

5. World Health Organization. Electronic recording and reporting for tuberculosis care and control. Geneva , Switzerland: World Health Organization, 2012.

6. World Health Organization. Digital health for the End TB Strategy: An agenda for action. Geneva, Switzerland: The End TB Strategy; European Respiratory Society;World Health Organization, 2015.

7. Islam MS, Banu S, Tarannum S, Chowdhury KIA, Nazneen A, Islam MT, et al. Examining patient management and healthcare workers exposures to pulmonary TB patients in two public tertiary care hospitals, Bangladesh. PLOS Global Public Health. 2021;In Press.

8. World Health Organization. WHO guidelines on tuberculosis infection prevention and control, 2019 update. Geneva, World Health Organization: World Health Organization, 2019 Contract No.: WHO/CDS/TB/2019.1.

9. Zetola NM, Macesic N, Modongo C, Shin S, Ncube R, Collman RG. Longer hospital stay is associated with higher rates of tuberculosis-related morbidity and mortality within 12 months after discharge in a referral hospital in Sub-Saharan Africa. BMC Infectious Diseases. 2014;14(1):409. doi: 10.1186/1471-2334-14-409.

10. Nardell EA. Transmission and Institutional Infection Control of Tuberculosis. Cold Spring Harbor perspectives in medicine. 2015;6(2):a018192-a. doi: 10.1101/cshperspect.a018192. PubMed PMID: 26292985.

11. Escombe AR, Oeser CC, Gilman RH, Navincopa M, Ticona E, Pan W, et al. Natural ventilation for the prevention of airborne contagion. PLoS medicine. 2007;4(2):e68. Epub 2007/03/01. doi: 10.1371/journal.pmed.0040068. PubMed PMID: 17326709; PubMed Central PMCID: PMCPMC1808096.

12. James Atkinson, Yves Chartier, Carmen Lúcia Pessoa-Silva, Paul Jensen, Li Y, Seto W-H. Natural Ventilation for Infection Control in Health-Care Settings. Geneva 27, Switzerland: World Health Organization, 2009 Contract No.: ISBN 978 92 4 154785 7 (NLM classification:WX 167).

13. Escombe AR, Ticona E, Chávez-Pérez V, Espinoza M, Moore DAJ. Improving natural ventilation in hospital waiting and consulting rooms to reduce nosocomial tuberculosis transmission risk in a low resource setting. BMC Infectious Diseases. 2019;19(1):88. doi: 10.1186/s12879-019-3717-9.

14. Zwama G, Diaconu K, Voce AS, O'May F, Grant AD, Kielmann K. Health system influences on the implementation of tuberculosis infection prevention and control at health facilities in low-income and middle-income countries: a scoping review. BMJ Global Health. 2021;6(5):e004735. doi: 10.1136/bmjgh-2020-004735.

15. World Health Organization. Implementing the WHO Policy on TB Infection Control in Health-Care Facilities, Congregate Settings and Households A framework to plan, implement and scale-up TB infection control activities at country, facility and community level WHO Policy on TB Infection Control in Health-Care Facilities, Congregate Settings and. Geneva: World Health Organization, The tuberculosis coalition for technical asistance, Centers for disease control and prevention, United States Agency International Development, 2010.

16. World Health Organization. WHO policy on TB infection control in healthcare facilitis, congregate settings and households. WHO,Geneva: WHO/HTM/TB/2009; 2009.

17. World Health Organization. Guidelines on Core Components of Infection Prevention and Control Programmes at the National and Acute Health Care Facility Level. Geneva: World Health Organization., 2016.

18. Sinuff T, Cook D, Giacomini M, Heyland D, Dodek P. Facilitating clinician adherence to guidelines in the intensive care unit: A multicenter, qualitative study. Critical care medicine. 2007;35(9):2083-9. Epub 2007/09/15. doi: 10.1097/01.ccm.0000281446.15342.74. PubMed PMID: 17855822.

19. Turnberg W, Daniell W, Simpson T, Van Buren J, Seixas N, Lipkin E, et al. Personal healthcare worker (HCW) and work-site characteristics that affect HCWs' use of respiratory-infection control measures in ambulatory healthcare settings. Infect Control Hosp Epidemiol. 2009;30(1):47-52. Epub 2008/12/03. doi: 10.1086/592707. PubMed PMID: 19046059.

20. Engelbrecht M, Rau A, Kigozi G, Janse van Rensburg A, Wouters E, Sommerland N, et al. Waiting to inhale: factors associated with healthcare workers’ fears of occupationally-acquired tuberculosis (TB). BMC Infectious Diseases. 2019;19(1):475. doi: 10.1186/s12879-019-4115-z.

---

## [Editor Report · Decision Letter 1]

13 Jan 2022

Preparedness of tertiary care hospitals to implement the National TB infection prevention and control guidelines in Bangladesh: A qualitative exploration

PONE-D-21-24921R1

Dear Dr.Islam,

I am pleased to inform you that your manuscript is accepted for publication in PLOS One. In your attempt to answer all the questions raised by the reviewers, you have revised your manuscript appropriately.

Kind regards,

Rabia Hussain

Academic Editor

PLOS ONE
---

## [Editor Report · Acceptance letter]

26 Jan 2022

PONE-D-21-24921R1 

Preparedness of tertiary care hospitals to implement the National TB infection prevention and control guidelines in Bangladesh: A qualitative exploration 

Dear Dr. Islam:

I'm pleased to inform you that your manuscript has been deemed suitable for publication in PLOS ONE. Congratulations! Your manuscript is now with our production department. 

Kind regards, 

on behalf of

Dr. Rabia Hussain 

Academic Editor

PLOS ONE